# Hybrid Chitosan Biosorbents: Tunable Adsorption at Surface and Micropore Domains

**DOI:** 10.3390/biomimetics9120725

**Published:** 2024-11-24

**Authors:** Inimfon A. Udoetok, Mohamed H. Mohamed, Lee D. Wilson

**Affiliations:** Department of Chemistry, University of Saskatchewan, 110 Science Place, Saskatoon, SK S7N 5C9, Canada; inimfon.udoetok@usask.ca (I.A.U.); mom133@mail.usask.ca (M.H.M.)

**Keywords:** chitosan, glutaraldehyde, surface quaternization, biopolymer cross-linking, adsorption, pore structure

## Abstract

Herein, we report a study that provides new insight on the knowledge gaps that relate to the role of biopolymer structure and adsorption properties for chitosan adsorbents that are cross-linked with glutaraldehyde. The systematic modification of chitosan cross-linked with glutaraldehyde (CG) and its quaternized forms (QCG) was studied in relation to the reaction conditions: mole ratios of reactants and pH conditions. Complementary adsorbent characterization employed ^13^C NMR/FTIR spectroscopy, TGA and DSC, point-zero-charge (PZC), solvent swelling, and sorption studies using selected dye probes. The spectral and thermal techniques provide complementary evidence that affirm the key role of cross-linker content and quaternization on variation of the physicochemical properties of chitosan. The PZC results reveal a neutral surface charge for the modified materials between pH 6.0 to 6.3 ± 0.3, as compared with pH 8.7 ± 0.4 for pristine chitosan. Solvent swelling in water decreased with greater cross-linking, while the QCG materials had greater swelling over CG materials due to enhanced hydration. The adsorption results reveal variable dye uptake properties according to the cross-linker content. Similarly, surface versus micropore adsorption was demonstrated, according to the nature and ionization state of the dye for the modified adsorbents, where the CG and QCG materials had tunable sorption properties that exceeded that of unmodified chitosan. A key step in tuning the structure and surface chemical properties of cross-linked chitosan involves pH control during synthesis. The facile tunability of the physicochemical properties of the modified biopolymers reported herein means that they possess features of biomimetics that are relevant to advanced drug delivery, antimicrobial materials for wound healing, biosensors, and biosorbents for biomedical applications.

## 1. Introduction

Chitosan (CH) is a copolymer that contains β-(1→4)-2-amino-2-deoxy-D-glucopyranose units derived from alkaline deacetylation of chitin [1,2,3,4]. After cellulose, chitosan is among the most abundant biopolymers. It bears a structural similarity to cellulose aside from the functionality at carbon-2 (-NHR; R is H or acetyl) of the glucopyranose monomer unit. Chitosan is unique due to the presence of amine groups, which confer water solubility at slightly acidic conditions. The synthetic versatility of chitosan as a key component in composites is reflected by synthetic modification of its hydroxyl and amine functional groups [5,6,7,8]. The physicochemical properties of chitosan can be modified via cross-linking of the free amine groups [9,10,11,12,13,14,15,16,17]. Cross-linked chitosan materials display wide-ranging applications that include biomedical devices [18,19], biosensors [20], bone scaffolds [17], and adsorbent materials [21,22,23,24,25,26,27,28,29,30].

One commonly used cross-linker employed in modifying the properties of chitosan is glutaraldehyde (GLU) [18,22,31,32]. Significant research has been directed at understanding the physicochemical properties of these materials by varying the cross-linker content (ratio of chitosan glucosamine unit and glutaraldehyde), pH, temperature, and concentration of acid media for dissolution of chitosan [32,33,34,35,36]. Since the seminal work by Kildeeva et al. [37] illustrated that the optimum condition for crosslinking chitosan with GLU was pH 5.6, follow-up studies [9,34,35,38,39,40,41] have reported the synthesis of cross-linked materials at acidic conditions, where no further pH adjustments are carried out in acidic media. As well, many studies reported that cross-linking of chitosan was carried out at elevated cross-linker levels, where the GLU:chitosan monomer mole ratio often exceeded unity and the mode of cross-linker addition was not fully described [9,32,33,34,35,36,38,39,40,41]. Hence, there are knowledge gaps related to structural aspects of cross-linked biopolymers and the physicochemical properties that relate to the synthetic conditions. This knowledge gap may be addressed by the systematic design and characterization of these materials by varying the cross-linker content, pH conditions, and the mode of reagent addition.

Recently, Quinlan et al. [42] developed a modified chitosan–glutaraldehyde cross-linked bead system by further quaternization of accessible –OH groups of chitosan with glycidyl trimethyl ammonium chloride (GTMAC) upon crosslinking. The resulting bead materials were employed in the sequestration of 2-naphthoxyacetic acid from aqueous solution. To our knowledge, this was the first example of such a quaternized chitosan–glutaraldehyde material. In view of the reported physicochemical properties of the materials for this system, it was concluded that comparatively few reports detail the structure–adsorption property relationships for these relatively simple materials.

Herein, the overall goal of this study was to achieve greater control over the textural properties and surface modification of chitosan by studying various synthetic parameters that govern the cross-linking reaction. To this end, the preparation and characterization of cross-linked chitosan and its quaternized forms were prepared at variable glutaraldehyde levels via rapid addition of the cross- at pH 5.6 under controlled conditions. In turn, the equilibrium adsorption properties of modified chitosan was studied by the use of model dyes with variable chemical structures and ionic (cationic, anionic) charges. The utility of these adsorptive probes to provide greater insight into the adsorption-based processes for these materials was shown, according to the role of glutaraldehyde-based cross-linking with chitosan, especially at levels below typical stoichiometric conditions. This study contributes to the design of improved chitosan materials in several ways: (i) the rate of glutaraldehyde addition to chitosan (and pH control) was shown to affect the physicochemical and adsorption properties of cross-linked chitosan, (ii) the role of quaternization of cross-linked chitosan on the surface adsorption properties, (iii) the variable role of micropore versus surface adsorption sites in cross-linked and quaternized chitosan and (iv) reduced levels of glutaraldehyde (below stoichiometric) are recommended for the effective modification of chitosan to optimize its pore structure and adsorption properties. The novelty of this study highlights the role of synthetic conditions (modality of cross-linker addition and mole ratio, surface quaternization, and pH conditions) on the structure and adsorption properties of modified chitosan. The results reported herein provide an understanding of the effects of step-wise cross-linking and quaternization of chitosan under controlled conditions, which are demonstrated to yield uniquely variable structure–property relationships. In terms of the adsorption properties, this study reports a *first example* of surface versus micropore adsorption that reveals the dependence of synthetic modification of the surface sites versus framework pore structure of the biopolymer network. The innovative use of dye probes reveals that surface versus micropore adsorption depend strongly upon the nature of the dye probe, pH conditions, cross-linker content and/or surface quaternization of chitosan as demonstrated herein. In the context of biomimetics [5,6,17,18,19], this study is envisaged to contribute to bioadsorption-based processes such as advanced drug delivery, antimicrobial biofilms, wound healing materials, biosensors, and bioadsorbent materials with antimicrobial properties for biomedical applications.

## 2. Results and Discussion

As outlined above, the role of controlled reaction conditions on the physicochemical and adsorption properties of cross-linked and quaternized chitosan has been sparsely reported [9,32,33,34,35,36,38,39,40,41]. Thus, the objective of the current study relates to the synthesis and characterization of cross-linked and quaternized chitosan, as outlined below. In turn, this study is anticipated to provide greater insight into the role of reaction conditions on the structure–function properties of modified chitosan with an emphasis on the equilibrium adsorption properties with several dye probe systems.

### 2.1. Characterization Results

#### 2.1.1. FTIR

The FT-IR spectra of chitosan and the modified forms (CG and QCG) are shown in Figure 1. The spectra display features that reflect the various functional groups of chitosan, along with groups due to the synthetic modifications of CG and QCG. The features include a broad band at ~3000–3700 cm^−1^, attributed to the stretching vibrations of O–H and N–H groups of chitosan. Other IR bands reflect C–H stretching vibrations (~2800–3000 cm^−1^), N–H bending bands (~1550–1640 cm^−1^) as well as C–O–H, C–O–C and C–N–H asymmetric stretching vibrations (~1000–1200 cm^−1^) [43,44]. The spectral features of the CG and QCG materials bear similar features to that of chitosan, revealing that the basic structural units of chitosan are preserved. However, the spectra of the modified materials display additional features apart from the IR spectra of chitosan, in agreement with reports on the effects of cross-linking and/or quaternization reactions [32,45,46,47]. These features include greater band broadening due to stretching vibrations of O–H groups and N–H groups (~3000–3700 cm^−1^), as well as the C–H groups (~2800–3000 cm^−1^). In Figure 1A,B, the attenuation of the NH_2_ deformation vibration at 1601 cm^−1^ is noted for the spectra of the cross-linked and cross-linked/quaternized materials, supporting the participation of such groups in the cross-linking and/or quaternization reaction [32,48].

Further evidence of glutaraldehyde cross-linking with chitosan concurs with the appearance of the new IR bands (~1717 cm^−1^ and ~1674 cm^−1^) assigned to the vibrational modes of aldehyde and imine groups. These signatures appear only for the spectra of the CG (*cf.* Figure 1A,C) and QCG (*cf.* Figure 1B) materials, where the spectral intensity increases according to greater glutaraldehyde content, in agreement with other reports [37,39]. The increase in the intensity of the band at ~1717 cm^−1^ with increasing GLU content suggests that cross-linking is favored at lower glutaraldehyde levels, whereas a combination of cross-linking and grafting is inferred at greater GLU content. A conspicuous difference between the CG and QCG materials at the lowest cross-linking ratio is the enhancement of the CH deformation band at ~1425 cm^−1^ for the CG polymer. The attenuation of IR intensity and greater band broadening for the modified materials occurs as the glutaraldehyde content increases. This is accounted for by a greater rigidity of the polymer network upon cross-linking and/or quaternization. Also, the apparent lack of notable differences in the spectral features of the cross-linked versus cross-linked + quaternized biopolymer at higher cross-linker ratios may also be related to the rigidity of the biopolymer network that restricts the vibrational modes of the groups introduced due to the cross-linking and quaternization steps.

#### 2.1.2. Thermal Analysis

The thermal stability profiles of chitosan and its modified forms (CG and QCG) are presented in Figure 2A. The profiles are presented as derivative (DTG) profiles, as shown by plots of derivative weight (wt.%/°C) versus temperature (°C). The DTG profiles reveal two thermal events between ~40 to 400 °C for chitosan and three events from ~40 to 480 °C for the modified forms, which concur with other reports [43,49]. The thermal events for pristine chitosan are characterized by the loss of bound water at ~50 °C and a decomposition event at ~300 °C. In Figure 2A, cross-linked chitosan (CG) shows a loss of bound water ~80 °C, degradation of the chitosan domains ~240 to 260 °C, followed by decomposition of the cross-linked sites near ~420 to 440 °C for the modified materials. Similarly, the QCG material (*cf.* Figure 2B) shows loss of bound water (~80 °C), decomposition of the chitosan domain (~260 to 270 °C) and degradation of the cross-linked/quaternized domain (~430 °C) for the different materials. The results show that the modified (CG and QCG) polymers display lower thermal stability relative to pristine chitosan, in parallel agreement with the alteration of the crystallinity of chitosan due to *pillaring* effects [43] of chitosan fibrils upon cross-linking and quaternization. This effect is supported by the attenuated FT-IR band intensities of the materials in Figure 1. An interesting feature of the DTG profiles for the materials is the correlation of the DTG intensity of the thermal event of the cross-linked domain to incremental cross-linker content. This trend supports the claim of exclusive cross-linking occurring at low cross-linker mole ratios, whereas concurrent cross-linking and grafting occur at higher cross-linker content for the CG materials.

Figure 2C illustrates the DSC profiles of chitosan and the modified materials (CG and QCG) between 40–180 °C. The endotherm for desorption of water occurs in a typical range, where the T_max_ and enthalpy (J/g) are shown in the parentheses, respectively: CH (97.4, 45.90), CH-0.125 (105.6, 138.5), QCG-0.125 (125.4, 147.2), CG-2 (112.7, 206.2), and QCG-2 (113.5, 216). The trend in T_max_ and enthalpy values vary as CH undergoes incremental cross-linking with GLU, where both parameters show an increase due to more favorable hydration of the modified CH materials versus CH. As expected, quaternization of the CG polymers had an increased enthalpic effect for incremental glutaraldehyde content, where the improved hydration was related to the presence of charged imine groups [50].

#### 2.1.3. ^13^C Solids NMR Spectroscopy

The ^13^C solids NMR spectra of chitosan and its modified forms are presented in Figure 3. The spectra of pristine chitosan display the following features attributed to the six carbon atoms (C1–C6) found in the monomer unit of chitosan: C1 (~105 ppm), C2 (~56.8 ppm), C3, 5 (~75.0 ppm), C4 (~82.4 ppm), C6 (~61.6 ppm), C=O (~174 ppm), and CH_3_ (~23.1 ppm), which finds support from other reported works [33,40,41]. Similarly, the ^13^C NMR spectra of CG (*cf.* Figure 3A) and QCG (*cf.* Figure 3B) bear similarities to chitosan and provide structural support that the framework structure of chitosan is preserved overall, in agreement with the FT-IR spectral and TGA results. However, the cross-linking and/or quaternization of chitosan is also affirmed by unique spectral features observed for CG and QCG. These features include spectral broadening, merging and induced chemical shifts of some ^13^C NMR lines for the modified materials. As well, the appearance of non-resolved spectral signals are more evident at incrementally higher cross-linker ratios. The new spectral signals include methylene groups of glutaraldehyde and GTMAC with different chemical environments at ~28.0–45.0 ppm, a conjugated ethylenic bond at ~100, 120–155 ppm and an imine bond at 174–175 ppm for the various polymers [32,37,46,51]. The intensity of these new NMR lines increase as the cross-linker mole ratio increases, in parallel with the incremental cross-linking of chitosan. A comparison of the spectra of CG and QCG polymers reveals that the C=O band for the CG polymers decrease as the cross-linker ratio increases, whereas the signature is evidently absent in the NMR spectra of QCG polymers. As the C2 band for CG polymers becomes merged with the C6 signals, its intensity for QCG polymers increases at a lower cross-linking ratio that undergoes attenuation as the cross-linker content increases. These effects are related to the introduction of C-N groups to the QCG polymers through the quaternization process.

#### 2.1.4. Point-of-Zero Charge (PZC)

The PZC of a sorbent material is the pH condition where the net surface charge of a material surface becomes zero [52,53]. For the case of adsorption studies, the PZC value provides insight on the electrostatic interactions of charged sorbate species with adsorbent surfaces [54,55]. The adsorbent surface is negatively charged when pH > pH_pzc_. Thus, an adsorptive attraction occurs with positively charged adsorbates. By contrast, negatively charged adsorbates are bound when pH < pH_pzc_. The PZC values for CH, CG and QCG materials was estimated by plotting the change in pH (final pH (pH_f_)-initial pH (pH_i_)) against pH_i_ (*cf*. Figure 4).

The pH_pzc_ of CH is 8.7± 0.4, while for CG-0.125 it is 6.0 ± 0.3 and for CG-0.5 and CG-2 it is 6.2 ± 0.3. The values for QCG-0.125, QCG-0.5 and QCG-2 are 6.0 ± 0.3, 6.2 ± 0.3 and 6.3 ± 0.3, respectively. The pH_pzc_ for CH concurs with other reported values [56,57,58,59]. Overall, the cross-linking of CH with GLU attenuates its pH_pzc_, in favorable agreement with other reports for cross-linking of CH with aldehydes, where the values ranged between 6.6–6.7 [60,61]. It can be inferred that surface effects will not be significant for sorption studies conducted at ambient conditions near pH 7, due to the proximity of the PZC for the various chitosan-based materials.

#### 2.1.5. Equilibrium Swelling Studies

The sorption of water by materials provides insight on the nature of the relative hydrophilic character and the textural properties of an adsorbent material [62]. As well, the propensity of composite adsorbents to undergo swelling relates to its ability to undergo significant structural changes due to hydration effects [63,64]. The equilibrium swelling properties of chitosan and its modified forms are presented in Figure 5.

The results show that CG2 had the least swelling (%) in water (83 ± 7%), in line with other reports [41,44]. The results further reveal that the solvent swelling (%) in water for CG and QCG decreased as the relative cross-linking ratio increased. Incremental cross-linking results in greater rigidity of the biopolymer framework that limits its propensity to expand upon solvent uptake, irrespective of the presence of micropores. The greater rigidity of the framework upon cross-linking concurs with NMR line broadening effects, as noted in the ^13^C NMR spectral results and by the reduced solvent swelling (%) reported herein. A comparison of the swelling (%) for CG and QCG shows that QCG exhibits greater swelling over CG, where a 100% increase was noted at the lowest cross-linker ratio. This trend may relate to the enhanced quaternization of chitosan, especially at a low cross-linking ratio (QCG-0.125) due to steric effects. Quaternization of chitosan results in electrostatic repulsion between the quaternary ammonium groups, along with enhanced hydration of the biopolymer due to the presence of charged surface sites [42]. Greater quaternization for QCG0.125 is affirmed by the enhanced ^13^C NMR spectral signal for C2 at ~56.8 ppm and the swelling (%) values reported herein. As the level of cross-linking increases, the accessibility for quaternization is anticipated to decrease, as evidenced by the downward trend in swelling from QCG-0.125 to QCG-2, in agreement with the reduced water uptake. The above results suggest that the solubility of the materials decreases with an increasing cross-linking ratio. The enhanced swelling properties of CG-0.125, and QCG-0.125 materials with a low cross-linking ratio relative to chitosan supports the variation in the hydration profile of the materials, where the enhanced swelling can facilitate gel and film-forming by these materials. Enhanced swelling ability is also envisaged to support drug loading into the hydrogel network to support drug delivery and wound healing applications.

### 2.2. Dye Adsorption Studies

#### 2.2.1. Decolorization of Methyl Orange (MO), Reactive Black 5 (RB), Phenolphthalein (Phth) and Methylene Blue (MB)

Figure 6A shows the relative decolorization efficiency of chitosan, CG and QCG polymers for methyl orange (MO), reactive black 5 (RB), phenolphthalein (Phth) and methylene blue (MB). All determinations were carried out at ambient pH (~6.2) except for Phth, which was obtained at alkaline conditions (pH 10.5). The removal efficiency of MO in its anion form by the chitosan materials showed the following trend: QCG2 > QCG 0.125 > CG2 > CG0.125 > CH. The degree of decolorization (dye removal) of MO upon binding with the biopolymers relate to the combined effects of cross-linking and quaternization on the adsorption process. Cross-linking may contribute defect sites and micropores within the chitosan framework, resulting in an incremental adsorbent surface area and cationic charge due to the presence of imine sites [28,43,53,65]. As well, quaternization increases the overall adsorbent surface charge [66]. The trend in dye decolorization suggests that electrostatic and hydrophobic contributions are key driving forces for MO adsorption by the polymers. The two-fold driving forces support that the adsorption of MO occurs at the surface and micropore domains of the polymers, in agreement with the pHpzc of the polymers as well as the similar uptake values for QCG-0.125 and QCG2. QCG-0.125 have a greater number of surface sites due to the dual role of imine and quaternary ammonium sites. As noted above, the extent of quaternization decreases as the degree of cross-linking increases due to inferred steric effects. QCG2 has more micropore sites due to the greater level of cross-linker content. Since MO exists in its anionic form at pH 6.2, favorable electrostatic interactions with the protonated imine sites and quaternary cations of modified chitosan occur, along with apolar interactions with the arene carbon sites of the dye [67,68,69,70,71,72,73,74,75].

In the case of RB, the removal efficiency of the various biopolymer systems adopt the following trend: QCG-0.125 > CH > CG-0.125 > QCG-2 > CG-2 (*cf.* Figure 6A). This trend reveals that elevated cross-linking of chitosan does not favor the adsorption of RB. However, the introduction of quaternary ammonium groups significantly enhanced the electrostatic interaction with the sulfonic groups of the RB dye. At a higher cross-linker mole ratio, the amine groups are consumed and are unavailable for RB dye adsorption. Filipkowska and Jóźwiak [76] also reported attenuated adsorption of RB by cross-linked chitosan, in agreement with the results herein. This study finds support for the primary role of electrostatic interactions in the adsorption process, while hydrophobic effects appear to be of secondary importance relative to the primary role of ion exchange [58,59,77]. Electrostatic effects can be inferred based on the pH_pzc_ of the biopolymers and the observed removal efficiencies: QCG-0.125, CH, CG-0.125, and QCG-0.125. These systems possess many surface adsorption sites (quaternary ammonium, protonated imine sites, and amine/ammonium groups) due to the minimal level of cross-linking and greater quaternization. Similarly, CH and CG-0.125 possess free amine or more accessible imine groups due to the effects of lower cross-linking.

The removal efficiency of (un)modified chitosan toward Phth occurs as follows: QCG-2 > QCG-0.125 > CG-2 > CG-0.125 > CH (*cf.* Figure 6A). The trend shows that hydrogen bonding, electrostatic interactions and hydrophobic effects can provide the driving force for Phth decolorization by the modified chitosan polymers. As well, the decolorization efficiency increased as the cross-linker content increased, in agreement with other reports [22,44,53]. CH displays the lowest decolorization efficiency due to its relatively hydrophilic nature, where binding of the dye solely occurs via hydrogen bonding. QCG-2 and QCG-0.125 display high removal efficiencies due to the availability of quaternary ammonium and micropore sites, along with -OH groups that afford dye decolorization via electrostatic and apolar interactions. The lower decolorization efficiency of Phth relative to MO and RB may relate to charge-screening effects since Phth occurs as a dianion at pH 10.5, whereas the surface charge of the biopolymers are negative, according to the pH_pzc_ results (*cf.* Figure 4).

The biopolymers display minimal removal efficiency with MB relative to the other dyes reported herein. The removal efficiency of MB by the materials adopted the following order: CH > CG-0.125 > CG-2 > QCG-2 > QCG-0.125. The minimal removal efficiency of the materials for MB relates to charge repulsion due to the formation of quaternary and protonated imine sites upon cross-linking or quaternization. MB is a cationic dye at pH 6.2, whereas the biopolymer surface is positively charged according to the pHpzc results (*cf.* Figure 4). The highest removal efficiency by CH relative to the modified materials may relate to (i) stronger n − π* interactions between the NH_2_ and OH groups of CH with the π* electrons of MB [30,64,78,79,80,81], and (ii) CH has the most negative electrostatic potential, which becomes attenuated upon further cross-linking or quaternization. The above effect is supported by the attenuation of the removal efficiency of MB for the CG and QCG biopolymers as the cross-linker content increases. Cross-linking of CH with GLU reduces the availability of the NH_2_ groups according to imine linkage formation, while quaternization attenuates the accessibility of the OH groups, in agreement with the lowest removal efficiency for the QCG polymers.

#### 2.2.2. Effects of pH on Methyl Orange Decolorization

The effects of pH on the decolorization efficiency of MO by chitosan and its hydrogels (CG and QCG) is presented in Figure 6B. The decolorization of MO at variable pH conditions adopted the following trend: pH 6.2 > pH 4.5 > pH 10.5. The observed profile may be explained by the speciation (charge state) of MO, along with the nature of the biopolymers at variable pH conditions. At pH 6.2, MO exists in its anion form, while the surface of the modified chitosan is neutral, according to the pH_pzc_ results in Figure 4. This implies that MO binds to the biopolymers via electrostatic interactions with the surface of chitosan at pH 6.2, along with adsorption via dispersion interactions within the apolar micropore domains of framework sites [70,71]. This is supported by the trend in the decolorization efficiency by the various systems: QCG-2 > QCG-0.125 > CG-2 > CG-0.125 > CH. Cross-linking of chitosan creates micropore domains while quaternization modifies the surface charge of the framework, as evidenced by the trend for the decolorization of MO at pH 6.2 [43,53]. MO is a zwitterion at pH 4.5 due to the protonation of one of the N-atoms. The zwitterion form of MO may account for its attenuated decolorization efficiency at pH 4.5 due to charge repulsion between the positive charge on the dye and the positively charged surface sites of the biopolymers. QCG-2 had the greatest decolorization efficiency while CH had none. This result provides support that the role of hydrophobic effects and electrostatic interactions [70,71] serve as key driving forces for dye decolorization, where the presence of favorable micropore domains in QCG-2 afforded the observed decolorization efficiency. Chitosan was not effective at pH 4.5 due to its greater solubility at this pH condition.

At pH 10.5, the biopolymers had the least decolorization efficiency due to the existence of MO in its anionic form. The surface charge of the biopolymer framework was either neutral from excess hydroxide in solution or had a negative surface charge due to ionization of the –OH groups of chitosan at alkaline pH [72,73,74,75]. QCG-2 displayed the greatest decolorization efficiency at this pH due to the role of apolar micropore domains and quaternary ammonium ions. CG-2 exhibited the second highest removal efficiency due to greater cross-linking that affords the presence of abundant micropore domains in the biopolymer framework. This trend further supports the role of hydrophobic effects and electrostatic interactions as the key driving forces described above, which concur with other studies on the adsorption properties of modified biopolymers [43,53,82].

### 2.3. Effects of pH Adjustment on Cross-Linking

The study reported by Kildeeva et al. [37] at pH 5.6 reported that the mobility of an α-hydrogen relative to the carbonyl group of glutaraldehyde plays a key role in the cross-linking of chitosan with GLU. In turn, the physicochemical and sorptive properties were evaluated for chitosan cross-linked without pH adjustment upon dissolution in acetic acid (pH 3.3; CG-0.125 B) and after pH adjustment to 5.6 (CG-0.125 A). The thermal degradation, swelling and adsorption properties of the biopolymers are presented in Figure 7.

The results in Figure 7A show that chitosan displays the lowest swelling in water relative to the cross-linked polymers. The swelling results also reveal that CG-0.125 A exhibits greater swelling than CG-0.125 B, illustrating that cross-linking at variable pH conditions affords variable *pillaring* effects [43,82] to the cross-linked polymers. The observed trend reveals that the swelling properties of CG polymers were greater at lower cross-linker content, in agreement with the swelling results reported in Section 2.1.5. The greater swelling at a lower cross-linker content may relate to the greater flexibility of the framework, where the role of pillaring effects of the chitosan biopolymer network allows for favorable hydration of the chitosan materials.

The thermal degradation profiles also support greater cross-linking for CG-0.125 A, according to the thermal event at ~420 to 440 °C (*cf.* Figure 7B), where CG-0.125 A has a greater peak area relative to CG-0.125 B, in agreement with greater levels of cross-linking reported elsewhere [22,38,43]. The thermal event at ~420 to 440 °C for chitosan cross-linked with glutaraldehyde was previously shown to be due to degradation of the cross-linked domains of the polymer, in agreement with the absence of such effects in the DTG profile for unmodified chitosan [64,81].

Similarly, the sorption properties of chitosan cross-linked at a very low GLU content (0.125 mole) also affirms that greater cross-linking occurs upon pH adjustment of the reaction mixture prior to the addition of GLU. MO and 2-naphthoxy acetic acid (2NAA) (*cf.* Figure 7C) were used as the dye probes. According to the results in Figure 7C, chitosan had the lowest removal efficiency, in line with the absence or negligible cation binding sites for the adsorbates. By contrast, CG-0.125 A and CG-0.125 B display different adsorbate removal efficiencies, according to the level of cross-linking of chitosan due to imine or quaternary cation binding sites. The removal efficiency results concur with the DTG and solvent swelling results (*cf.* Figure 7B) reported herein along with other reports [43,44,53,64]. The above results provide support for greater cross-linking of chitosan with glutaraldehyde due to the role of pH, in agreement with the seminal study by Kildeeva et al. [37].

## 3. Experimental

### 3.1. Materials

Chitosan with ~80% deacetylation was obtained from the University of Saskatchewan (Saskatoon, Canada). 2-Naphthoxy acetic acid (2NAA), sodium hydroxide, glutaraldehyde (GLU), reactive black 5 (RB), phenolphthalein (Phth), methylene blue (MB), methyl orange (MO) and glycidyl trimethyl ammonium chloride (GTMAC) were obtained from Sigma–Aldrich Canada Ltd. (Oakville, ON, Canada). ACS grade glacial acetic acid was obtained from EMD chemicals (New Jersey, United States of America). All materials were used as received without further purification. For all quantitative experiments, the results were obtained in triplicate measurements with an estimated standard error in the range of ±(2–5)%.

### 3.2. Synthesis of Cross-Linked Chitosan

The synthesis of modified biopolymer materials (CGx and QCGx, where x represents the respective mole content of glutaraldehyde: 0.125, 0.25, 0.5, 1 and 2; cf. Table 1, Figure 1A,B) was adapted from literature [38,39,43,83] with minor modifications. A complementary illustration of the micropore domains is presented elsewhere (cf. Figure 1 in [43]). Briefly, ~2.50 g of chitosan was stirred ca. 960 rpm in 200 mL (5% *v*/*v* glacial acetic acid) in a beaker until completely dissolved. The pH of the chitosan solution was adjusted to 5.6 before the addition of the requisite volumes of GLU that were added rapidly whilst stirring. Gelation was achieved within a few minutes upon glutaraldehyde addition, and then the reaction was continued for 12 h. The gel was broken through the addition of a 1 M NaOH solution with vigorous magnetic stirring until the pH of the mixture reached pH 7, affording precipitation of the product. The product was separated from the supernatant through vacuum filtration while washing copiously with cold Millipore water. A similar batch of polymer was synthesized at the lowest cross-linker mole ratio by the above procedure without any pH adjustment. For the quaternized polymers (QCGx), the cross-linked (CGx) polymers were added to 100 mL of Millipore water in a round bottom flask at 65 °C, followed by the addition of 4.14 mL of GTMAC, where the reaction was run at 65 °C for 12 h (*cf*. Figure 1A). The quaternized materials were separated from the supernatant by vacuum filtration. All products were washed with cold Millipore water for the complete removal of unreacted GLU and/or GTMAC, as confirmed by UV–Vis spectrophotometric analysis (Varian CARY 100, Varian Australia Pty Ltd., Mulgrave, Australia) of the supernatant. The synthesized materials were dried in a vacuum oven at 60 °C for 12 h. The products were ground with a mortar and pestle and sieved using a 40-mesh sieve.

### 3.3. Characterization

#### 3.3.1. FTIR Spectroscopy

The FTIR spectra of chitosan and its modified forms (CG and QCG) were obtained using a Bio-RAD FTS-40 IR spectrophotometer (Bio-Rad Laboratories, Inc., Philadelphia, PA, USA). Preparation of the samples involved grinding with a mortar and pestle to a fine powder. The powdered samples were analyzed in neat form with the spectrophotometer set to reflectance (Diffuse Reflectance Infrared Fourier Transform; DRIFT) mode with a resolution of 4 cm^−1^. 512 scans were recorded over the 400–4000 cm^−1^ spectral range at 295 K.

#### 3.3.2. Thermal Gravimetry Analysis (TGA)

The thermal stability of chitosan and the cross-linked (CG) and quaternized (QCG) polymers were evaluated using a TA Instruments Q50 TGA system (New Castle, DE, USA) operated under the following conditions: heating rate −5 °C min^−1^ up to 500 °C with a N_2_ carrier gas. The results are displayed as first derivative (DTG) plots of weight against temperature (wt.%/°C) versus temperature (°C).

#### 3.3.3. Differential Scanning Calorimetry (DSC) Studies

The DSC profiles of the chitosan materials were acquired using a TA Q20 thermal analyzer (New Castle, DE, USA), as follows: scan rate (5 °C min^−1^) with dry N_2_ gas for temperature regulation and purging of the sample compartment. The air-dried samples (ca. 5 mg) were sealed hermetically in aluminum pans and analyzed by heating over the range of 50–180 °C. The results are reported as a plot of heat flow (W/g) versus temperature (°C).

#### 3.3.4. pH at the Point-of-Zero Charge (pHpzc)

The pH where the surface of the materials has a net zero surface charge (pHpzc) was determined, as described by Singh et al. [84]. Briefly, 0.01 M stock solution of NaCl was prepared and the pH of several (5 × 50 mL) portions were adjusted between pH 2 to 10 using NaOH/HCl such that each portion had a different pH. Each solution (25 mL) was transferred into five 30 mL vials containing 50 mg of the modified chitosan. The vials containing the electrolyte + sample were agitated in a horizontal shaker at 295 K for about 48 h. The final pH was measured using a Mettler Toledo pH meter, where a graph of change in pH vs initial pH was plotted, and the pHpzc recorded as the point where the difference (pH,final − pH,initial = 0) was zero.

#### 3.3.5. Equilibrium Swelling Properties in Water

The equilibrium swelling properties of chitosan and the modified biopolymers were determined by adding approximately 50 mg of the sample and 30 mL of Millipore water into 36 mL vials. The sample/water system was agitated in a horizontal shaker for ~48 h. The swollen weight of the sample (W_s_) was determined after tamping dry with filter paper, where the dry weight (W_d_) was obtained upon drying at 60 °C to a constant weight. The swelling ratio was calculated by Equation (1):(1)$$Sw(\%)=\frac{Ws-Wd}{Wd}\times 100$$

#### 3.3.6. ^13^C Solids NMR Spectroscopy

^13^C solids NMR spectra of chitosan materials were acquired with a Bruker AVANCE III HD spectrometer Analyzer (PerkinElmer, Inc., Waltham, MA, USA). The spectrometer is equipped with a 4 mm DOTY CP-MAS (cross polarization with magic angle spinning; CP-MAS) solids probe operating at 125.77 MHz (^1^H NMR frequency = 500.23 MHz). The ^13^C NMR CP/MAS acquisition conditions include the following: spinning speed (10 kHz), ^1^H-90 pulse (3.5 µs), contact time (0.75 ms) and a ramp pulse on the ^1^H channel. Variable scans (n = 2000) were accumulated with a recycle delay (2 s), where all experiments were recorded using a 71 kHz SPINAL-64 decoupling sequence during acquisition. Adamantane was used as the external standard (δ = 38.48 ppm for the low field signal).

### 3.4. Sorption Studies

The adsorption properties of chitosan and its modified forms was evaluated using several dyes: reactive black 5 (RB), phenolphthalein (Phth), methylene blue (MB), and methyl orange (MO). Dye solutions (1 mM) were prepared at ambient pH for RB, MO and MB with Phth and another MO prepared at pH 10.5 using a 0.1 M NaHCO_3_ [44]. A fixed volume (7 mL) of the different dye solutions were added to each 10 mL vial containing ca. 10 mg of sample. The samples were equilibrated at 22 °C on a horizontal shaker table for 24 h, where the dye concentration before (C_o_) and after sorption (C_e_) was determined using a double beam spectrophotometer (Varian CARY 100, Varian Australia Pty Ltd., Mulgrave, Australia) at 22 ± 0.5 °C. The dye removal efficiency for the various chitosan adsorbents was calculated by Equation (2):(2)$$\%\;Removal=\frac{Co-Ce}{Co}\times 100$$

## 4. Conclusions

Chitosan was incrementally cross-linked with glutaraldehyde (GLU) along with quaternization using glycidyl trimethyl ammonium chloride (GTMAC). The cross-linked (CG) and cross-linked/quaternized (QCG) polymers were characterized through complementary spectral (^13^C NMR and FTIR), thermal (TGA and DSC), PZC, solvent swelling and dye adsorption studies. The characterization results provide support for the cross-linking of chitosan with GLU to yield imine sites, including the quaternization process to generate quaternary ammonium cations. Incremental cross-linking (and imine linkages) develops as the GLU content increases, where the IR and NMR spectral results show an increase related to the imine bond signatures. The sorption and solvent swelling results also reveal variable swelling and adsorption properties for the biopolymers according to the cross-linker content, where surface versus micropore adsorption (*cf*. Figure 1 in [43]) was demonstrated according to the chemical nature of the dye. The superior pore structure and surface properties of the CG and QCG polymers were evidenced by their unique dye decolorization efficiencies relative to pristine chitosan. Furthermore, pH control of the chitosan solution before the cross-linking reaction affects the physicochemical and adsorption properties of the resulting cross-linked polymer, where pH 5.6 was optimal for the formation of imine linkages with chitosan to yield superior properties. This work is a *first example* of detailed systematic study of chitosan and its cross-linked forms along with quaternization of its surface sites. We address the knowledge gaps that relate to the structure and adsorption properties of such modified chitosan materials [82]. The results herein have provided further insight into strategies to tailor porous and surface functionalized adsorbents to address the removal of anion target species from aqueous media. The surface quaternization of such adsorbents have significant potential as biomimetics for adsorption-based processes such as targeted drug delivery, antimicrobial biofilms, wound healing materials, and adsorbents with antimicrobial properties for biomedical applications [5,6,17,85,86,87].

## Data Availability

The raw data supporting the conclusions of this article will be made available by the authors upon reasonable request.

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
