# Peer review of "Hybrid Chitosan Biosorbents: Tunable Adsorption at Surface and Micropore Domains"

_biomimetics, 2024, doi:10.3390/biomimetics9120725_

Round 1

Reviewer 1 Report

Comments and Suggestions for Authors

In the study, the authors reported the synthesis and characterization of hybrid chitosan biosorbents, and investigated the adsorption for the dye. However, several necessary questions need to be made in the manuscript.

1. BET and SEM of the adsorbents need to be characterized.

2. The effect of contact time and initial concentration on adsorption needs to be investigated.

3. The relationship of micropore domains and adsorption needs to be discussed.

4. The probable adsorption mechanism should be provided.

5. The authors should check the format of all references.

Author Response

Response to Reviewer Comments on the Manuscript ID: biomimetics-3285639 - Hybrid Chitosan Biosorbents: Tunable Adsorption at Surface and Micropore Domains

Reviewer 1

Comments and Suggestions for Authors

In the study, the authors reported the synthesis and characterization of hybrid chitosan biosorbents and investigated the adsorption for the dye. However, several necessary questions need to be made in the manuscript.

  1. BET and SEM of the adsorbents need to be characterized.

Response: We thank the reviewer for this comment. The authors agree that BET and SEM characterization would be very valuable for manuscript. However, the main aim of the study is to elucidate the relationship between polymer structure and adsorption properties for chitosan adsorbents that are cross-linked with glutaraldehyde. The authors believe that structural characterization techniques and the use of dye probes for characterization of surface vs micropore adsorption of the materials provided the information required to proof the authors hypotheses.

  1. The effect of contact time and initial concentration on adsorption needs to be investigated.

Response: We thank the reviewer for this comment. The authors agree that investigating the effect of contact time and initial concentration on adsorption would be beneficial, especially in the case of kinetics of adsorption. Since the focus of the current study was related to equilibrium dye adsorption, an investigation of contact time is outside the scope of the present study, which was to elucidate the relationship between polymer structure and equilibrium adsorption properties for chitosan adsorbents that are cross-linked with glutaraldehyde.

  1. The relationship of micropore domains and adsorption needs to be discussed.

Response: We thank the reviewer for this comment. The relationship between the micropore domains and adsorption was included in the revised manuscript.

  1. The probable adsorption mechanism should be provided.

Response: We thank the reviewer for this comment. The probable mechanism of adsorption is now included in the revised manuscript. Please refer to section 2.2 where electrostatic, hydrophobic, and n - π* interactions along with hydrogen bonding were identified as contributing interactions in the adsorption process.

  1. The authors should check the format of all references.

Response: We thank the reviewer for this comment. All references have been revised according to the recommended format

Reviewer 2 Report

Comments and Suggestions for Authors

The manuscript presents a comprehensive study on the incremental cross-linking of chitosan with glutaraldehyde, followed by quaternization using GTMAC. The authors use a broad range of techniques for the comprehensive characterization of the chitosan modifications and properties of the modified polymers, including spectral (FTIR, 13C NMR) and thermal methods (TGA, DSC), Point-of-zero charge (PZC) and equilibrium swelling studies, as well as dye adsorption studies. The results support the formation of imine sites due to cross-linking and quaternary ammonium cations during quaternization, contributing to unique adsorption properties. The study connects the degree of cross-linking and quaternization with the sorption capacity and swelling behavior and highlight the importance of pH control in the synthesis, with pH 5.6 being optimal for forming imine linkages. The study outlines the potential applications of quaternized and cross-linked chitosan in biomedical fields such as drug delivery, antimicrobial biofilms, and wound healing. 

However, several key aspects could be further explored to strengthen the manuscript:

1. The manuscript would benefit from the explanation of how the varying degrees of cross-linking and quaternization affect the solubility of the resulting polymers, as well as its gel- and film-forming properties, as these are critical features for applications in wound healing and drug delivery.

2. Considering the significant chemical and physicochemical changes observed in the biopolymer structure and confirmed by FTIR and 13C NMR, the study would also benefit from morphological analysis (e.g., SEM or AFM) to confirm changes in surface topography and porosity. This would help correlate the structural changes with adsorption behavior.

3. If the authors aim to recommend these polymers for biomedical applications, it is essential to assess their toxicity profile. Has this aspect been previously studied, or do the authors plan to perform such study in the future?

4. The modifications to chitosan described in the study offer enhanced adsorption properties and surface functionalization. Could the authors elaborate on how these modifications would make the use of chitosan more convenient or advantageous for specific applications, such as drug delivery, antimicrobial surfaces, or water purification? A more detailed explanation would highlight the broader impact of this work.

5. Minor corrections: line 275 "at" should be "as"

Author Response

Response to Reviewer Comments on the Manuscript ID: biomimetics-3285639 - Hybrid Chitosan Biosorbents: Tunable Adsorption at Surface and Micropore Domains.

Reviewer 2

The manuscript presents a comprehensive study on the incremental cross-linking of chitosan with glutaraldehyde, followed by quaternization using GTMAC. The authors use a broad range of techniques for the comprehensive characterization of the chitosan modifications and properties of the modified polymers, including spectral (FTIR, 13C NMR) and thermal methods (TGA, DSC), Point-of-zero charge (PZC) and equilibrium swelling studies, as well as dye adsorption studies. The results support the formation of imine sites due to cross-linking and quaternary ammonium cations during quaternization, contributing to unique adsorption properties. The study connects the degree of cross-linking and quaternization with the sorption capacity and swelling behavior and highlight the importance of pH control in the synthesis, with pH 5.6 being optimal for forming imine linkages. The study outlines the potential applications of quaternized and cross-linked chitosan in biomedical fields such as drug delivery, antimicrobial biofilms, and wound healing. 

However, several key aspects could be further explored to strengthen the manuscript:

  1. The manuscript would benefit from the explanation of how the varying degrees of cross-linking and quaternization affect the solubility of the resulting polymers, as well as its gel- and film-forming properties, as these are critical features for applications in wound healing and drug delivery.

Response: We thank the reviewer for this comment. The discussion on the effect of varying degrees of cross-linking and quaternization on the solubility of the materials is included in the revised manuscript. Please refer to lines 258 – 263 of the revised manuscript.

  1. Considering the significant chemical and physicochemical changes observed in the biopolymer structure and confirmed by FTIR and 13C NMR, the study would also benefit from morphological analysis (e.g., SEM or AFM) to confirm changes in surface topography and porosity. This would help correlate the structural changes with adsorption behavior.

Response: We thank the reviewer for this comment. The authors agree that BET and SEM characterization would be very valuable for manuscript. A challenge related to the interpretation of the BET and SEM results relates to measurements carried out in the absence of a solvent medium as outlined elsewhere (Appl. Sci. 202414, 7577. doi: 10.3390/app14177577). The main aim of the study is to elucidate the relationship between polymer structure and adsorption properties for chitosan adsorbents that are cross-linked with glutaraldehyde. The authors believe that structural characterization techniques employed and the use of dye probes for characterization of surface vs micropores adsorption of the materials provided the information required to address the objectives of this study.

  1. If the authors aim to recommend these polymers for biomedical applications, it is essential to assess their toxicity profile. Has this aspect been previously studied, or do the authors plan to perform such study in the future?

Response: We thank the reviewer for this comment. The main goal of the current study is to elucidate the relationship between polymer structure and adsorption properties for chitosan adsorbents that are cross-linked with glutaraldehyde. The toxicity profile of these materials will be studied as part of future research.

  1. The modifications to chitosan described in the study offer enhanced adsorption properties and surface functionalization. Could the authors elaborate on how these modifications would make the use of chitosan more convenient or advantageous for specific applications, such as drug delivery, antimicrobial surfaces, or water purification? A more detailed explanation would highlight the broader impact of this work.

Response: We thank the reviewer for this comment. The effectiveness of the modification of chitosan through cross-linking and quaternization in making chitosan more convenient or advantageous for drug delivery and water purification is included in the revised manuscript. Please refer to lines 226 – 228 and 258 – 263 of the revised manuscript.

  1. Minor corrections: line 275 "at" should be "as"

Response: We thank the reviewer for this comment. The error has been corrected as recommended.

Reviewer 3 Report

Comments and Suggestions for Authors

The manuscript is dedicated to preparation of cross-linked chitosan and study of its properties as adsorbent of dyes. The cross-linking was carried out with glutaraldehyde. Additionally the glycidyl trimethyl ammonium chloride was used to obtain ionic form of the obtained cross-linked polymer. There are some notices, need to be done to improve the manuscript.

1. Table 1. Please provide in text before the Table a prior definition for the abbreviation GTMAC.

2. It is recommended to provide a chemical scheme for preparation of cross-linked polymers. 

3. Line 131. "Further evidence of glutaradehyde cross-linking with chitosan concurs with the appearance of the new IR bands (∼1717 cm–1 and ∼1674 cm–1)..." it is hard to see these peaks on Figure 1. I recommend to provide magnified region of the spectra, where these signals will be more distinctive.

4. It would be more logical to replace the paragraph of Thermal analysis after all the spectral analyses.

5. It is recommended for better understanding to provide integral forms of TGA curves, not only differential in Figure 2.

6. In some places it is written ВЕП and in some TGA. Please choose one style for all the text.

7. Lines 188-190. What exactly mean C1, C2, C3 etc.?

8. Line 200.  "conjugated ethylenic bond at ∼100..." how exactly the conjugated ethylenic bond could be formed. According to the chemical structures of the reagents and products, there cannot be conjugated bonds.

9. Authors claim about microporous structure of the polymers but there was no BET nitrogen adsorption analyses. So, how it was proved, that the polymers contain micropores?

Author Response

Response to Reviewer Comments on the Manuscript ID: biomimetics-3285639 - Hybrid Chitosan Biosorbents: Tunable Adsorption at Surface and Micropore Domains

Reviewer 3

The manuscript is dedicated to preparation of cross-linked chitosan and study of its properties as adsorbent of dyes. The cross-linking was carried out with glutaraldehyde. Additionally the glycidyl trimethyl ammonium chloride was used to obtain ionic form of the obtained cross-linked polymer. There are some notices, need to be done to improve the manuscript.

  1. Table 1. Please provide in text before the Table a prior definition for the abbreviation GTMAC.

Response: The authors are grateful to the reviewer for taking time to review the article. The position of Table 1 has been moved and the definition of GTMAC is included in the revised manuscript.

  1. It is recommended to provide a chemical scheme for preparation of cross-linked polymers.

Response: We thank the reviewer for the comment. A schematic description of the synthetic procedure has ben included as recommended.

  1. Line 131. "Further evidence of glutaradehyde cross-linking with chitosan concurs with the appearance of the new IR bands (∼1717 cm–1 and ∼1674 cm–1)..." it is hard to see these peaks on Figure 1. I recommend to provide magnified region of the spectra, where these signals will be more distinctive.

Response: We thank the reviewer for this comment. A spectra showing 1660 – 1800 cm-1 is included in the revised manuscript.

  1. It would be more logical to replace the paragraph of Thermal analysis after all the spectral analyses.

Response: We thank the reviewer for this comment. The authors believe that placing the  figure at the current position provides easy reference for the reader.

  1. It is recommended for better understanding to provide integral forms of TGA curves, not only differential in Figure 2.

Response: We thank the reviewer for this comment, the authors believe that the form of TGA curves presented in the manuscript provided the best evidence in terms of onset temperatures and temperature at the maximum weight loss events. The authors contend presentation of the integral forms do not likely provide additional visual information relative to the DTG profile.

  1. In some places it is written ВЕП and in some TGA. Please choose one style for all the text.

Response: We thank the reviewer for the comments. TGA was used when referring to the analysis while DTG is used when referring to the results.

  1. Lines 188-190. What exactly mean C1, C2, C3 etc.?

Response: We thank the reviewer for this comment. The sentence has been revised to provide understanding for what C1 to C6 represents.

  1. Line 200. "conjugated ethylenic bond at ∼100..." how exactly the conjugated ethylenic bond could be formed. According to the chemical structures of the reagents and products, there cannot be conjugated bonds.

Response: We thank the reviewer for this comment. However, the formation of this bond has been previously reported based on the mechanism of reaction. See the following papers:

  1. https://doi.org/10.1016/S0141-8130(99)00068-9
  2. Surface Grafted Chitosan Gels. Part II. Gel Formation and Characterization | Langmuir
  3. Authors claim about microporous structure of the polymers but there was no BET nitrogen adsorption analyses. So, how it was proved, that the polymers contain micropores?

Response: We thank the reviewer for this comment. The presence of micropores in glutaraldehyde crosslinked chitosan materials has already been established by previous studies. See the following studies:

  1. Engineering Porosity-Tuned Chitosan Beads: Balancing Porosity, Kinetics, and Mechanical Integrity | ACS Omega
  2. https://doi.org/10.1016/j.ijbiomac.2023.124373.

Therefore, authors relied on already established information for chitosan/glutaraldehyde materials in their discussion.

Round 2

Reviewer 1 Report

Comments and Suggestions for Authors

After considering the changes made by the authors, I would like to recommend this revised manuscript for publication.

Author Response

Response to Reviewer Comments on the Manuscript ID: biomimetics-3285639 - Hybrid Chitosan Biosorbents: Tunable Adsorption at Surface and Micropore Domains.

We appreciate the valuable comments provided by the reviewer and we have presented responses accordingly in a point-by-point fashion, as outlined in green font.

Reviewer #1

After considering the changes made by the authors, I would like to recommend this revised manuscript for publication.

Response:

We appreciate the constructive comments and valuable insight provided by Reviewer #1, along with the dedicated effort and time in reviewing this manuscript.

Reviewer 3 Report

Comments and Suggestions for Authors

Unfortunately authors haven't fully respondend nad made revisions according to my notices.

For notice #2 i asked to provide chemical structures and chemical reactions between functional groups.

Notice #4. It is logical that first the chemical structure is studied and then thermal and other properties are studied.

Notice #5. The integral TGA curve should be given to see not only the degradation temperatures but also to see the weight loss and char yield.

Notice #7 For this notice it is also better to provide chemical structure where the C1-C6 should be marked.

Author Response

Response to Reviewer Comments on the Manuscript ID: biomimetics-3285639 - Hybrid Chitosan Biosorbents: Tunable Adsorption at Surface and Micropore Domains.

We appreciate the valuable comments provided by the reviewer and we have presented responses accordingly in a point-by-point fashion, as outlined in green font.

For notice #2 i asked to provide chemical structures and chemical reactions between functional groups.

Response: We thank the reviewer for this comment. The chemical structure has been included as recommended. See scheme 1B.

Notice #4. It is logical that first the chemical structure is studied and then thermal and other properties are studied.

Response: We thank the reviewer for the comment. However, the authors believe that the manuscript reads well based on how it is structured since the TGA results can be treated independently of detailed structural information. We contend that the ordering of TGA and structural results is because discussion was structured in a step-wise fashion, where the results presented in the preceding sections provide support for those in the sections that follow.

Notice #5. The integral TGA curve should be given to see not only the degradation temperatures but also to see the weight loss and char yield.

Response: We thank the reviewer for this comment, the authors believe that the form of TGA curves presented in the manuscript provided the best evidence in terms of onset temperatures and temperature at the maximum weight loss events. The authors contend that presentation of the integral forms that show the weight loss and char yield and do not provide additional visual information relative to the DTG profile in the context of the weight loss events.

Notice #7 For this notice it is also better to provide chemical structure where the C1-C6 should be marked.

Response: We thank the reviewer for this comment. The chemical structure has been included as recommended, as illustrated in Scheme 1B.
